# Valorization of Coffee Cherry Pulp into Potential Functional Poultry Feed Additives by Pectinolytic Yeast *Kluyveromyces marxianus* ST5

**DOI:** 10.3390/ani15152311

**Published:** 2025-08-07

**Authors:** Thanongsak Chaiyaso, Kamon Yakul, Wilasinee Jirarat, Wanaporn Tapingkae, Orranee Srinual, Hien Van Doan, Pornchai Rachtanapun

**Affiliations:** 1Division of Biotechnology, Faculty of Agro-Industry, Chiang Mai University, Chiang Mai 50100, Thailand; kamon.y@cmu.ac.th (K.Y.); jirarat4824@gmail.com (W.J.); 2Center of Excellence in Agro Bio-Circular-Green Industry (Agro BCG), Faculty of Agro-Industry, Chiang Mai University, 155 Moo 2, Mae Hia, Muang, Chiang Mai 50100, Thailand; pornchai.r@cmu.ac.th; 3Department of Animal and Aquatic Sciences, Faculty of Agriculture, Chiang Mai University, Chiang Mai 50200, Thailand; wanaporn.t@cmu.ac.th (W.T.); orranee.s@cmu.ac.th (O.S.); hien.d@cmu.ac.th (H.V.D.); 4Functional Feed Innovation Center (FuncFeed), Faculty of Agriculture, Chiang Mai University, Chiang Mai 50200, Thailand; 5Division of Packaging Technology, Faculty of Agro-Industry, Chiang Mai University, Chiang Mai 50100, Thailand

**Keywords:** coffee cherry pulp, *Kluyveromyces marxianus* ST5, antioxidant, in vitro digestion, bioaccessibility

## Abstract

Coffee cherry pulp (CCP) is a nutrient-rich by-product that is often discarded in the coffee industry. This study used *Kluyveromyces marxianus* ST5 to ferment CCP, enhancing its value as a poultry feed additive. The yeast produced pectin-degrading enzymes, improving antioxidant activity and nutrient availability. Optimized fermentation increased enzyme production and antioxidant levels. Simulated digestion tests showed higher peptide release and antioxidant activity in fermented CCP. These findings suggest that fermented CCP may support poultry health, offering a sustainable and low-cost alternative to conventional feed additives and reducing agro-industrial waste.

## 1. Introduction

Thailand is the third-largest coffee producer in Asia, with approximately 11,000 tons of Arabica coffee produced annually in the northern region [1,2]. During green bean processing, by-products such as skin, pulp, mucilage, and parchment account for about 55% of the total coffee weight, representing a significant source of agro-industrial waste [3,4]. Among these, coffee cherry pulp (CCP), the outer skin and mucilage of the coffee fruit removed during bean processing, is often discarded despite its rich composition of carbohydrates (~50%), protein (~10%), fiber (~20%), fat (~2.5%), caffeine (~1.3%), and phenolic compounds [3,4,5]. Due to its high polyphenol and antioxidant contents, CCP has recently gained attention as a potential bioresource rather than waste [3].

Recent studies have highlighted its potential for value-added applications [4,5,6,7,8]. One promising approach to valorize CCP is fermentation, which enhances its antioxidative compounds [9]. This process depends on the growth and metabolic activity of various microorganisms, including yeasts, filamentous fungi, lactic acid bacteria (LAB), and acetic acid bacteria (AAB) [10]. Yeasts have become an effective microorganism for enhancing various agricultural by-products’ bioavailability and nutritional value, particularly CCP. Among them, pectinolytic yeasts such as *Wickerhamomyces anomalus*, *Pichia kudriavzevii*, *Saccharomyces cerevisiae* [10], and *K. marxianus* [11] effectively convert the complex compounds in coffee cherries into simple, more bioactive molecules. This fermentation process increases the total polyphenol and flavonoid contents, thereby boosting the antioxidant activity of the CCP [12]. *K. marxianus* is generally recognized as safe (GRAS), and exhibits robust pectinolytic activity, thermotolerance, rapid growth, and broad substrate utilization, making it suitable for agro-industrial bioconversion [13]. Moreover, improved antioxidant properties are crucial, as they can reduce oxidative stress when incorporated into animal diets. Additionally, the fermentation of pectinolytic yeasts helps to reduce anti-nutritional compounds, particularly tannins and phytates, found in raw CCP, making it a safer and more beneficial feed additive [14,15,16].

Pectinases are responsible for breaking down pectin substances and have significant importance in the food industry and are used as exogenous enzymes in feed additives [17]. The main enzymes involved in coffee fermentation are pectin lyase (PL) and polygalacturonase (PG) [14]. These enzymes have the potential to complete the digestion of pectin to produce galacturonic acid and its oligomers [18,19,20]. Numerous studies have demonstrated that fermentation significantly enhances antioxidant activity and polyphenolic content in CCP. For instance, fermentation with *Aspergillus tamarii* has a significantly higher total phenolic content (TPC) of 137.8 g/kg, compared with unfermented coffee pulp (122.0 g/kg) [21]. Haile and Kang [5] reported an increase in the antioxidant activity of fermented green coffee beans by selected yeasts, with the TPC of the extracts ranging from 1.11 to 1.30 GAE mg/mL, significantly higher than the unfermented value of 0.72 GAE mg/mL. Additionally, fermented CCP has potential as a functional feed ingredient due to its high antioxidant content, improved digestibility, and enhanced nutritional composition. For example, coffee pulp fermented with *A. niger* has been shown to be safely included in broiler diets at levels of up to 10–15% without negatively affecting weight gain or feed conversion, whereas unfermented pulp above 10% resulted in adverse effects [22,23].

Unlike previous studies that employed filamentous fungi and various yeast species for coffee pulp fermentation, the present study focused on the isolation of a novel pectinolytic yeast strain with potential applications in poultry nutrition. Therefore, this study aimed to screen and isolate pectinolytic yeasts for the bioconversion of CCP, focusing on *K. marxianus* due to its recognized safety and high enzymatic efficiency, aiming to enhance antioxidant activity and nutritional quality. The production of PL and PG was optimized using a central composite design (CCD). In addition, the in vitro bioavailability of fermented CCP was evaluated under simulated conditions of the chicken gastrointestinal tract. These findings provide a sustainable strategy for valorizing coffee by-products and support the development of functional feed additives to promote animal health.

## 2. Materials and Methods

### 2.1. Chemical and Medium Composition

All reagents used were of analytical grade. Polygalacturonic acid, 2,2-diphenyl-1-picrylhydrazyl (DPPH), 2,2-azino-bis(3-ethylbenzothiazoline-6-sulfonic acid) (ABTS), 2,4,6-tris(2-pyridyl)-s-triazine (TPTZ), pepsin, and pancreatin were purchased from Sigma-Aldrich (Steinheim, Germany). Citrus pectin was bought from Himedia (Nashik, India). Dried CCP was kindly provided by the Department of Animal and Aquatic Science, the Faculty of Agriculture, Chiang Mai University. The pectin enrichment medium was composed of citric pectin (2.0 g/L), MnSO_4_ (0.05 g/L), KH_2_PO_4_ (0.2 g/L), (NH_4_)_2_SO_4_ (1.0 g/L), CaCl_2_·2H_2_O (0.05 g/L), MgSO_4_·7H_2_O (0.8 g/L), and yeast extract (1.0 g/L). In contrast, the CCP medium consisted of 10% (*w*/*v*) or 100 g/L dried coffee cherry, 4.0 g/L (NH_4_)_2_SO_4_, 1.0 g/L (NH_4_)_2_HPO_4_, and 0.1 g/L MgSO_4_·7H_2_O, as modified from Haile and Kang [10]. The initial pH of both media was adjusted to 6.5 using 6 N NaOH. The media were sterilized at 121 °C for 15 min in an autoclave (Hirayama, Hiclave HVA-85, Tokyo, Japan).

### 2.2. Screening, Isolation, and Identification of Pectinolytic Yeast

An amount of 1 g of CCP, obtained from the coffee-processing plant of the Khun Chang Kien Royal Project, Chiang Mai, Thailand, was added to 50 mL of sterile pectin enrichment medium and incubated at 30 °C for 48 h under static conditions to enrich the pectinolytic yeasts. After incubation, the culture was serially diluted with 0.85% NaCl and spread onto pectin agar plates supplemented with 200 ppm chloramphenicol to suppress bacterial growth. Pectinolytic activity was detected by flooding the plates with a 50 mM potassium iodide–iodine solution; clear yellow halos around colonies indicated enzymatic pectin degradation [10]. To induce the production of PL and PG, 10% (*v*/*v*) of the starter of selected isolates was inoculated into 50 mL of CCP medium and cultured at 30 °C for 48 h under static conditions. The crude enzyme was then harvested by centrifugation at 10,000 rpm (9500× *g*) at 4 °C for 15 min using a refrigerated centrifuge (Z236K, Hermle Labortechnik, Wehingen, Germany). The cell-free supernatant was used to assay PL and PG activities, as well as antioxidant activity.

For molecular identification, yeasts were grown in YM broth for 24 h, and yeast cells were collected by centrifugation at 10,000 rpm (9500× *g*) and 4 °C for 15 min. The selected strain was identified based on its genotype using 26S rDNA gene sequence analysis. Briefly, genomic DNA from the selected strain was extracted using the GeneJET Genomic DNA Purification Kit (ThermoScientific, Waltham, MA, United States) and used as a template for constructing the 26S rDNA gene sequence. The 26S rDNA gene was amplified through polymerase chain reaction (PCR). The PCR reaction mixture consisted of the universal forward primer (5′-ACCCGCTGAACTTAAGC-3′), the reverse primer (5′-TACTACCACCAAGATCT-3′), and DreamTaq Green PCR Master Mix (2x) (ThermoScientific, Waltham, MA, United States). Afterward, the samples were sequenced by the DNA-sequencing service of ATGC Co., Ltd. (Khlong Nueng, Thailand) The DNA sequences were compared with those available in the GenBank database using the Basic Local Alignment Search Tool (BLAST) algorithm from the National Center for Biotechnology Information (NCBI). A phylogenetic tree was constructed using the neighbor-joining method for this alignment with the Molecular Evolutionary Genetics Analysis (MEGA 4, PSU, University Park, PA, United States) software.

### 2.3. Optimization of Pectinase Production by Central Composite Design (CCD)

The response surface methodology (RSM) using a central composite design (CCD) was employed to determine the optimal cultivation conditions and the individual and interactive effects of each factor on the production of PL and PG by the selected yeast strain. The pH (X1), CCP (X2), inoculum size (X3), and temperature (X4) were tested through submerged fermentation in a 250 mL Erlenmeyer flask containing 50 mL of CCP medium. All flasks were cultivated at 30 °C for 48 h under static conditions. The experiments were designed and analyzed by using the Stat-Ease software (Design-Expert 6.0.10, Stat-Ease Corporation, Minneapolis, MN, USA). Each factor was tested at five levels: −α, −1, 0, +1, and +α (Table 1). The results of the CCD are presented as a second-order polynomial using a multiple regression technique according to the following equation:Y = β_0_ + ∑β_i_x_i_ + ∑β_ii_x_i_^2^ + ∑β_ij_x_i_x_j_
where Y represents PL and PG activity (U/mL), β_0_ the intercept term, β_i_ the linear coefficients, β_ii_ the quadratic coefficients, β_ij_ the interactive coefficients, and x_i_ and x_j_ the coded independent factors.

### 2.4. Enhancement of Antioxidant Activity of Cherry Coffee Pulp Under Optimal Conditions

To enhance the antioxidant activity, CCP was used as a substrate. The starter culture of the selected yeast strain was prepared in YM medium and incubated at 30 °C and 200 rpm for 48 h. Following validation of the conditions obtained from the CCD, the optimized conditions were applied in a 2.0 L Duran bottle with a working volume of 1.5 L. The reaction mixture was autoclaved at 121 °C for 15 min. After sterilization, the suitable starter culture was inoculated into the reaction mixture. The fermentation was carried out under optimized conditions for 48–60 h under static conditions. The cell-free supernatant samples were then harvested by centrifugation at 10,000 rpm (9500× *g*) at 4 °C for 15 min and were used to assay PL and PG activities, as well as antioxidant activity. The unfermented CCP (without inoculation) was used as the control. Meanwhile, the whole fermented CCP was blended and then subjected to freeze drying (Labconco, FreezeZone 6, Kansas City, MO, USA) to obtain the fermented CCP powder, which was stored at 4 °C. The powder samples were used for antioxidant assays and polyphenol analysis by high-performance liquid chromatography (HPLC), with unfermented CCP powder serving as a control.

### 2.5. In Vitro Digestibility Assay Mimicking Chicken Digestive Tract

The unfermented and fermented CCP powders were evaluated for digestibility and bioavailability using an in vitro chicken digestion model based on a modified method of Bryan et al. [24]. For the gastric-phase digestion, a 500 mg sample (N × 6.25) was placed in a 50 mL centrifuge tube along with 50 mg of guar gum and 13.5 mL of a 50 mmol/L HCl solution. The pH of the mixture was adjusted to 2.5 using 6 N HCl, followed by vortexing. After thorough mixing, 1.5 mL of pepsin (30,000 U/mL) was added to initiate the reaction. The reaction mixture was incubated in a water bath at 41 °C for 30 min to simulate gastric digestion. For the intestinal digestion phase, the pH of the reaction mixture was adjusted to 7.5 by adding 4.9 mol/L NaOH immediately after the gastric phase. Subsequently, 2 mL of pancreatin (60,000 U/mL) was added to the reaction mixture. The reactions were then incubated in a water bath at 41 °C for 180 min to simulate intestinal digestion. Samples were collected at 0, 30, 60, 120, and 180 min. Then, each sample was centrifuged at 10,000 rpm (9500× *g*) for 5 min. The supernatant was then heated at 100 °C for 10 min to terminate the reaction. All samples were analyzed for their peptide concentration, DPPH and ABTS scavenging activities, ferric-reducing antioxidant power (FRAP), and TPC.

### 2.6. Analytical Method

#### 2.6.1. Pectin Lyase (PL) Activity

PL activity was determined using a modified method of Haile and Kang [10]. The assay involved mixing 5 mL of 1.0% (*w*/*v*) citrus pectin (85% esterified) with 0.5 M Tris–HCl buffer at pH 6.8 and 1.0 mL of cell-free culture supernatant. The reaction mixture was incubated for 2 h at 40 °C. To stop the reaction, 0.6 mL of 9.0% (*w*/*v*) zinc sulfate and 0.6 mL of 0.5 M sodium hydroxide were added. Subsequently, 5 mL of the reaction mixture was centrifuged at 6000 g (4430× *g*) at 4 °C for 5 min. The resulting supernatant was then combined with 3 mL of 0.04 M thiobarbituric acid, 2.5 mL of 0.1 M hydrochloric acid, and 0.5 mL of distilled water. This final mixture was boiled for 30 min and cooled to room temperature, and the absorbance was measured at 550 nm using a spectrophotometer (10S UV–Vis, Thermo Fisher Scientific, Waltham, MA, USA). An amount of 1 unit of PL activity (U/mL) was defined as an increase in absorbance by 0.01 unit.

#### 2.6.2. Polygalacturonase (PG) Activity

PG activity was determined according to the method of Haile and Kang [10]. The 0.1% (*w*/*v*) solution of polygalacturonic acid was prepared in 0.1 M citrate buffer at pH 5.0. The reaction mixture consisted of 1.0 mL of the polygalacturonic acid solution and 1.5 mL of the crude enzyme samples. This mixture was incubated at 40 °C for 1 h. The reaction was stopped by adding 1.5 mL of DNS (3,5-dinitrosalicylic acid reagent). The mixture was then boiled for 5 min and subsequently cooled in an ice bath. The concentration of released galacturonic acid was measured by monitoring the absorbance at 540 nm (A_540_) using a spectrophotometer. An amount of 1 unit of enzyme activity (U/mL) was defined as the amount of enzyme that liberated 1 mol of galacturonic acid per minute under the assay conditions.

#### 2.6.3. Antioxidant Activity

The antioxidant activity was determined using DPPH and ABTS radical-scavenging activities and the FRAP assay, with Trolox serving as the standard. The DPPH radical-scavenging activity was measured as described by Veenashri and Muralikrishna [25], with some modifications. Briefly, 0.50 mL of the sample was mixed with an ethanolic DPPH solution (80 mg/L) and incubated at 37 °C for 60 min. After incubation, the absorbance was measured at 517 nm. The ABTS radical-scavenging activity was assessed using a modified method by Shazly et al. [26]. A working solution of ABTS radicals was prepared by mixing a 2.45 mmol/L potassium persulfate solution with a 7 mmol/L ABTS solution in a 1:1 (*v*/*v*) ratio and incubating the mixture in the dark for 12–16 h. The absorbance of the ABTS solution at 734 nm was adjusted to 0.700 ± 0.02 using a 50 mmol/L potassium phosphate buffer (pH 7.4). The reaction mixture, containing 0.50 mL of the sample and 0.85 mL of the prepared ABTS solution, was incubated at 37 °C in the dark for 30 min. The radical-scavenging activity was determined by measuring the decrease in absorbance at 734 nm. The FRAP assay was performed according to the method described by Benzie and Strain [27]. The stock solution was prepared by mixing 0.3 mol/L acetate buffer (pH 3.6), 20 mmol/L FeCl_3_·6H_2_O, and 10 mmol/L TPTZ in 40 mmol/L HCl in a 10:1:1 (*v*/*v*/*v*) ratio. A reaction mixture was prepared by adding 0.1 mL of the sample to 0.9 mL of the FRAP reagent. This mixture was incubated at 37 °C in the dark for 30 min, and the absorbance was measured at 595 nm.

#### 2.6.4. Determination of Total Phenolic Content (TPC)

The TPC in each sample was measured using the Folin–Ciocalteu colorimetric method [28]. Briefly, 0.3 mL of the sample was combined with 1.5 mL of 0.2 M Folin–Ciocalteu phenol reagent. After allowing the mixture to incubate for 5 min, 1.2 mL of a 0.7 M sodium carbonate (Na_2_CO_3_) solution was added, and the solution was mixed thoroughly. The reaction tubes were then incubated at 37 °C in the dark for 2 h. The absorbance at 760 nm was measured using a spectrophotometer. The TPC of the samples was quantified using a calibration curve prepared with gallic acid, and the results were expressed as µmole of gallic acid equivalents per milliliter (µmole GAE/mL).

#### 2.6.5. HPLC Analysis of Phenolic Compounds

To investigate the presence of gallic acid (GA), chlorogenic acid (CA), caffeine (CaF), *p*-coumaric acid (*p*CM), and sinapic acid (SN), the CCP samples were extracted and analyzed using the HPLC-DAD method reported by Atlabachew et al. [29]. A 0.4 g sample of CCP powder was extracted with 60% methanol in an ultrasonic bath. A portion of the extract was treated with 20% lead acetate. The supernatant was filtered using a nylon membrane syringe filter and subjected to HPLC analysis. A 5.0 μL aliquot of the sample was injected into a reversed-phase C8 column (Supelco, 150 × 4.6 mm, 5 μm, Bellefonte, PA, USA), which was maintained at 25 °C. The separation was performed under isocratic conditions using a mobile phase of 90% acidified water (0.1% orthophosphoric acid) and 10% acetonitrile at a flow rate of 0.4 mL/min. Detection was realized at 272 nm for CaF and at 330 nm for all other standards, respectively.

#### 2.6.6. Statistical Data Analysis

All experiments were performed in triplicate. The data’s normality was confirmed before ANOVA, and all factors were considered fixed effects. The data were analyzed for statistical significance using one-way analysis of variance (ANOVA) followed by Duncan’s multiple-range test (*p* < 0.05). The statistical software SPSS v.20 (SPSS Inc., Chicago, IL, USA) was used to analyze the experimental data. For optimization studies, regression analysis and a response surface methodology based on a central composite design (CCD) were performed using Design-Expert version 6.0.10 (Stat-Ease Inc., Minneapolis, MN, USA), with model terms considered significant at *p* < 0.05.

## 3. Results

### 3.1. Screening and Isolation of Pectinolytic Yeasts

Based on the large, clear zone around colonies on the pectin agar plate, thirty-two isolates were screened from CCP samples. The ability of all isolates to produce PL and PG was confirmed. The results showed that only 11 isolates could grow and produce both PL and PG after cultivation in the CCP medium (Table 2). The PL and PG activities ranged from 2.07 to 3.74 U/mL and 3.19 to 6.32 U/mL, respectively. Notably, the maximal PL activity (3.74 ± 0.40 U/mL) was obtained from isolate C5, while the highest PG activity (6.32 ± 0.13 U/mL) was produced by isolate ST5. Furthermore, isolates with high enzyme activity demonstrated a significant ability to enhance the antioxidant activity of fermented coffee cherries. Isolate C5 exhibited antioxidant activity, with DPPH, ABTS, and FRAP values of 7168 ± 154, 5712 ± 159, and 3772 ± 210 µmole TE/L, and a TPC of 7574 ± 178 µmole GAE/L. Similarly, isolate ST5 demonstrated DPPH, ABTS, and FRAP values of 7231 ± 111, 5856 ± 135, and 4305 ± 141 µmole TE/L, as well as a TPC of 7680 ± 136 µmole GAE/L. These results suggest a correlation between high enzyme activity and increased antioxidant activity. Additionally, the 26S rDNA sequences of isolates C5 and ST5 showed high similarity to *Cyberlindnera fabianii* (KY108793.1) and *K. marxianus* (MT507790.1), with 99% identity. Unfortunately, *C. fabianii* is known as an opportunistic pathogen that is frequently isolated from clinical specimens [30], which renders it unsuitable for further study. Therefore, *K. marxianus* ST5 was selected for further research.

### 3.2. Optimization of Pectinases Production byCentral Composite Design (CCD)

The optimal levels of four factors, namely, initial pH (X_1_), CCP (X_2_), inoculum size (X_3_), and temperature (X_4_), and their effects on PL and PG production by *K. marxianus* ST5 were determined. The ANOVA results for the CCD are presented in Table 3 and Table 4. Moreover, the CCD constructed a quadratic model for PL and PG production (Y) as the following equation (coded factors):Pectin lyase (U/mL) = −35.68 + 5.20 (X_1_) + 1.38 (X_2_) + 2.35 (X_3_) + 0.99 (X_4_) − 0.45 (X_1_)^2^ − 0.05 (X_2_)^2^ − 0.08 (X_3_)^2^ − 0.02 (X_4_)^2^ − 0.005 (X_1_) (X_2_) − 0.09 (X_1_) (X_3_) + 0.005 (X_1_) (X_4_) − 0.05 (X_2_) (X_3_) + 0.024763 (X_2_) (X_4_) + 0.006 (X_3_) (X_4_)Polygalacturonase (U/mL) = −25.47 + 4.88 (X_1_) + 1.32 (X_2_) + 2.03 (X_3_) + 0.90 (X_4_) − 0.42 (X_1_)^2^ − 0.04 (X_2_)^2^ − 0.07 (X_3_)^2^ − 0.02 (X_4_)^2^ + 0.0007 (X_1_) (X_2_) − 0.10 (X_1_) (X_3_) + 0.004 (X_1_) (X_4_) − 0.05 (X_2_) (X_3_) + 0.02 (X_2_) (X_4_) + 0.007 (X_3_) (X_4_)

As depicted in Table 3 and Table 4, the R^2^ values of 0.9521 for PL and 0.9379 for PG indicate a high consistency between the actual and predicted values, with only 4.79% and 6.21% of the total variation not explained by the model. In this case, *p*-values of less than 0.05 indicated significant model terms and values. The terms X_1_, X_2_, X_3_, and X_4_ had a significant effect on PL and PG production. The quadratic coefficients, including X_1_^2^, X_2_^2^, X_3_^2^, and X_4_^2^, were also significant. The PL and PG activities ranged from 4.12 to 10.50 U/mL and 11.50 to 17.45 U/mL, respectively. The maximal PL and PG activities of 9.15 and 15.78 U/mL were obtained at pH 5.0, a CCP concentration of 15% (*w*/*v*) or 150 g/L, an inoculum size of 6% (*v*/*v*), and 25 °C. Moreover, the interaction between the concentration of CCP and inoculum size (X_2_X_3_) and the interaction between CCP concentration and temperature (X_2_X_4_) were significant for both PL and PG production, as illustrated in the 3D interactive plot (Figure 1).

Based on these results, the optimal conditions for PL and PG production were determined to be a CCP concentration of 16.81% (*w*/*v*) or 168 g/L, an inoculum size of 5.87% (*v*/*v*), an incubation temperature of 30 °C, and an initial pH of the medium of 5.24. Under these optimized conditions, the highest enzyme activities were predicted to be 9.52 U/mL for PL and 16.17 U/mL for PG. However, the validation experiments under the predicted conditions resulted in mean experimental values for PL and PG production of 9.17 ± 0.20 and 15.78 ± 0.14 U/mL. In addition, the antioxidant activities of the fermented CCP supernatant as determined by DPPH, ABTS, and FRAP assays were 8645 ± 215, 9871 ± 180, and 10,121 ± 235 µmole TE/L, respectively, while the TPC was 8068 ± 170 µmole GAE/L.

### 3.3. Antioxidant Profile and Content of Phenolic Compounds of Ccp Powder

Due to the complexity of the bioactive compound profiles in plants, mixed analytical procedures are typically employed to estimate the antioxidant capacity of samples. In this study, measurements of DPPH, ABTS, FRAP, and TPC were conducted to investigate the effect of yeast fermentation on CCP (Table 5). The fermented CCP powder exhibited significantly higher antioxidant activity, with values of 8545.92 ± 241 µmole TE/L for DPPH, 10,495.56 ± 192 µmole TE/L for ABTS, and 10,496.78 ± 176 µmole TE/L for FRAP, as well as a TPC of 8266.50 ± 154 µmole GAE/L. Meanwhile, unfermented CCP powder exhibited significantly lower antioxidant activity, with values of 4290.82 ± 262 µmole TE/L for DPPH, 5573.33 ± 180 µmole TE/L for ABTS, and 5336.96 ± 287 µmole TE/L for FRAP, as well as a TPC of 3236.11 ± 191 µmole GAE/L. Moreover, the polyphenols and caffeine in the CCP samples were estimated based on the results of five phenolic substances, which were gallic acid (GA), chlorogenic acid (CA), caffeine (CaF), *p*-coumaric acid (*p*CM), and sinapic acid (SN), as shown in Table 5. The high content of polyphenols was obtained from fermented cherry coffee powder, with values of 215.83 ± 1.44 µg/g for GA, 512.35 ± 1.56 µg/g for CA, 8.85 ± 1.02 mg/mL for CaF, and 22.88 ± 0.32 µg/g for SN. Overall, most assay results correlated with the antioxidant assays, as fermented CCP exhibited higher antioxidant activity than unfermented cherry coffee powders. However, the *p*CM content in fermented CCP powder was 3.64 ± 0.19 µg/g, which decreased from 5.92 ± 0.24 µg/g in the unfermented cherry coffee sample.

### 3.4. Bioaccessibility Under In Vitro Digestibility

The in vitro digestion model enabled the assessment of the digestibility and bioavailability of bioactive compounds in yeast-fermented CCP powder under conditions that mimic the chicken digestive system. The peptide concentration and antioxidant activity values for unfermented and fermented CCP powders are shown in Figure 2. The results showed that the fermented sample had a significantly higher peptide concentration and antioxidant activity compared with the unfermented sample. At 180 min, the highest values observed for the fermented CCP were a peptide concentration of 5.02 ± 0.17 mg/mL, a DPPH level of 5328 ± 280 μmole TE/L, an ABTS level of 12,826 ± 116 μmole TE/L, a FRAP level of 5895 ± 139 μmole TE/L, and a TPC of 6179 ± 163 μmole GAE/L. In contrast, the unfermented sample showed lower values, with a peptide concentration of 3.02 ± 0.11 mg/mL, a DPPH level of 4024 ± 61 μmole TE/L, an ABTS level of 8843 ± 143 μmole TE/L, a FRAP level of 4142 ± 74 μmole TE/L, and a TPC of 5560 ± 33 μmole GAE/L.

## 4. Discussion

Several yeast isolates (Table 2) showed a wide range of PL and PG activities, with high standard deviations in some cases, particularly in the PL activity of isolates C1 and C8. This variation may have resulted from common fermentation factors, such as substrate composition, inoculum size, pH, and temperature. Similar patterns have been observed in other studies on spontaneous fermentation systems [31]. Although this variability is typical in initial screening stages, it was addressed in later experiments by selecting the most consistent and high-performing isolate, *K. marxianus* ST5, for further optimization under controlled conditions. Isolate ST5 exhibited strong pectinolytic activity by producing PL and PG, enzymes essential for breaking down pectic substances in plant matrices [10,18]. Fermentation with this strain significantly enhanced the antioxidant properties of CCP, consistent with prior findings that microbial fermentation improves polyphenol availability and antioxidant capacity. For example, Luo et al. [32] reported that pectinase treatment of mulberry juice significantly increased chlorogenic acid and cyanidin-3-*O*-glucoside by 77.8% and 44.5%, respectively.

The capacity of isolate ST5 to utilize CCP as a carbon source and produce high levels of pectinolytic enzymes highlights its bioconversion potential. However, the initial enzyme yields were not appropriate for industrial applications. A CCD was applied to improve production, and CCP concentration, inoculum size, and temperature were identified as key factors, resulting in a 2.5–2.8-fold increase in PL and PG activities. These parameters play vital roles in microbial enzyme biosynthesis. CCP concentration affects substrate availability for growth and enzyme production [33], inoculum size determines microbial competitiveness [34], and temperature influences growth and enzyme synthesis [35]. Importantly, enhanced enzymatic activity under optimized conditions was associated with increased antioxidant capacity, likely due to the synergistic effects of enzymatic release of bound polyphenols from the plant matrix and the production of antioxidant metabolites by *K. marxianus* through its versatile metabolic pathways [36]. These compounds may act synergistically with polyphenols to enhance the overall antioxidant activity during fermentation. Haile et al. [37] observed improved antioxidant activity in fermented coffee beans after yeast fermentation. DPPH activity increased from 25.50 to 32.73%, and FRAP values increased from 172.50 to 176.11% after fermentation with *W. anomalus* KNU18Y3. Compared with previous studies using *Aspergillus niger*, which produced PL activities of 2.5–3.0 U/mL under solid-state fermentation [23], *K. marxianus* ST5 yielded 9.17 U/mL, indicating significantly higher enzymatic potential under submerged fermentation conditions. Moreover, the antioxidant activity observed in fermented CCP by isolate ST5 (DPPH 99.2%) exceeded that of fermented coffee by *W. anomalus* (DPPH ~32.7%) reported by Haile et al. [37], suggesting superior bioconversion efficiency.

After 48 h of fermentation with *K. marxianus* ST5, both the antioxidant activity and phenolic content in the fermented CCP powder significantly increased (Table 5), likely due to enzymatic hydrolysis that released bound phenolics and antioxidants from the plant matrix, thereby enhancing their bioavailability [38]. The elevated DPPH, ABTS, and FRAP values supported this trend. The fermentation also altered specific phenolic and caffeine profiles; GA, CA, CaF, and SN increased, while *p*CM decreased, suggesting active bioconversion [39]. As previously reported, these compounds are known for their antioxidant and anti-inflammatory properties [40,41]. Compared with other phytogenic additives used in poultry feed, such as ginger, black cumin, moringa, cinnamon, rosemary, chicory, coffee pulp, and coffee bean [42,43,44,45,46,47], fermented CCP might offer comparable functional benefits, including improved nutrient bioaccessibility and antioxidant potential.

The in vitro digestion model showed that fermented CCP had 1.2–1.9-fold higher peptide, phenolic, and antioxidant bioaccessibility than the unfermented sample. This reflects the action of pectinolytic enzymes, PL and PG, in enhancing nutrient release [18,46]. Biotechnological processes like enzymatic fermentation improve bioavailability by liberating bound phytochemicals [48,49]. Although the improved antioxidant capacity and phenolic release suggest potential applications in functional animal feed, particularly for poultry, caution is warranted. The findings are based solely on in vitro assays and simulated gastrointestinal conditions. While the results indicate the enhanced bioaccessibility of nutrients and antioxidants, they do not confirm animal physiological effects. This study was limited to in vitro conditions and lab-scale fermentation. The absence of in vivo trials means that absorption, metabolic fate, and systemic effects in poultry remain unknown.

Additionally, to contextualize the antioxidant potential of fermented CCP as a feed additive, it is essential to compare it with other plant-derived additives that have shown antioxidant activity in vivo. For instance, *Sulla flexuosa* and bitter vetch (*Vicia ervilia*) have been explored as alternative feed resources rich in bioactive compounds, contributing to improved growth performance and meat quality in livestock [50,51]. A similar report by Boukrouh et al. [52] demonstrated that supplementation with bitter vetch and sorghum grains improved carcass traits and fatty acid profiles in goats. Although these examples focus on leguminous plants, they highlight the broader feasibility of using antioxidant-rich agricultural by-products in animal nutrition. Nevertheless, the antioxidant efficacy of fermented CCP and its physiological benefits, bioavailability, and safety must still be validated through in vivo studies in poultry.

## 5. Conclusions

This study demonstrated that *K. marxianus* ST5 can effectively enhance the antioxidant properties and TPC of CCP by producing pectin-degrading enzymes. Optimization using a CCD significantly improved PL and PG activities (up to 9.17 and 15.78 U/mL, respectively), correlated with increased antioxidant activities and the release of key phenolic compounds. Fermentation with *K. marxianus* also improved the in vitro bioaccessibility of nutrients in CCP, suggesting its potential as a functional feed additive. While these findings demonstrate promising biochemical improvements, the present study was limited to in vitro models. Future research should focus on scaling up the fermentation process and conducting in vivo feeding trials to assess the impacts on poultry growth performance, oxidative status, and overall health. Furthermore, evaluations of product safety, palatability, and sensory acceptance are necessary to support practical implementation in livestock feeding systems.

## Figures and Tables

**Figure 1 animals-15-02311-f001:**
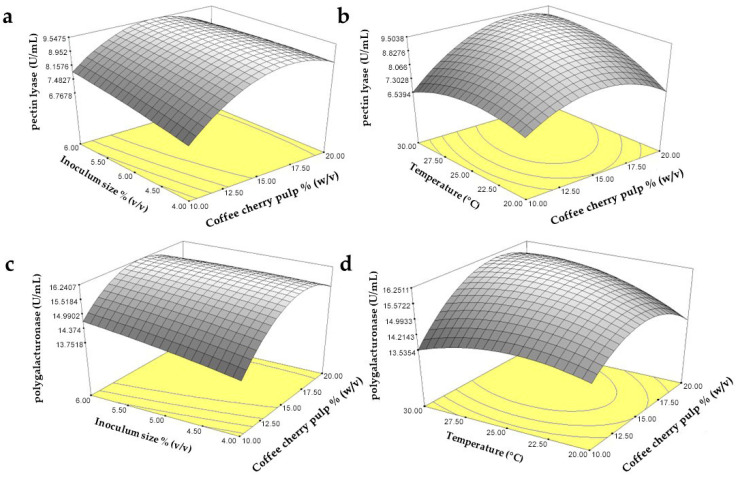
The 3D interactive plots of the interactions of inoculum size (X_2_) and coffee cherry pulp (X_3_) for PL (**a**) and PG (**c**), and coffee cherry pulp (X_3_) and temperature (X_4_) for PL (**b**) and PG (**d**).

**Figure 2 animals-15-02311-f002:**
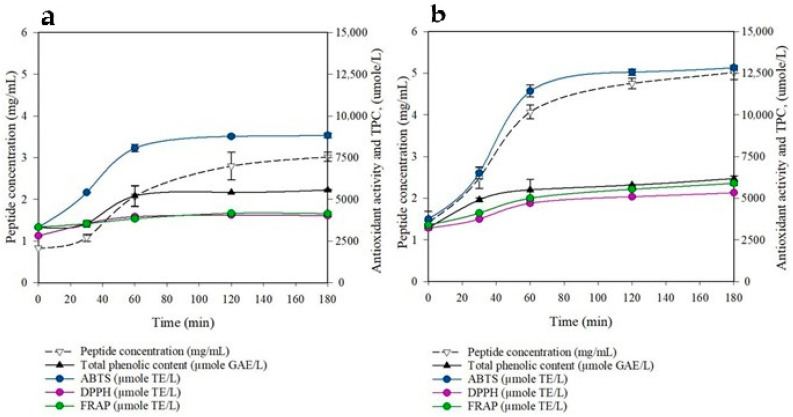
Changes in peptide concentration, antioxidant activity, and total phenolic content (TPC) during in vitro chicken digestion of (**a**) unfermented and (**b**) fermented CCP powders.

**Table 1 animals-15-02311-t001:** Symbol codes and levels of four factors studied using CCD in terms of actual and coded factors.

Factor	Symbol	Unit	Levels				
			−α	(−1)	(0)	(+1)	+α
pH	(X_1_)	-	2.0	3.5	5.0	6.5	8.0
Coffee cherry pulp (CCP)	(X_2_)	% (*w*/*v*)	5.0	10	15	20	25
Inoculum size	(X_3_)	% (*v*/*v*)	2.0	4.0	6.0	8.0	10
Temperature	(X_4_)	°C	15	20	25	30	35

**Table 2 animals-15-02311-t002:** Pectin lyase (PL), polygalacturonase (PG), antioxidant activity, and total phenolic content (TPC) values of 11 isolates in CCP medium at 30 °C for 48 h.

Isolate	Enzyme Activity (U/mL)		Antioxidant Activity (µmole TE/L)			TPC (µmole GAE/L)
	Pectin Lyase	Polygalacturonase	DPPH	ABTS	FRAP	
C1	2.07 ± 0.90 ^d^	3.52 ± 0.23 ^c^	6706 ± 180 ^b^	4339 ± 141 ^c^	3460 ± 113 ^b^	5385 ± 141 ^d^
C2	2.07 ± 0.43 ^d^	4.75 ± 0.25 ^b^	6090 ± 144 ^c^	5130 ± 195 ^b^	2846 ± 147 ^b^	6988 ± 129 ^b^
C3	2.25 ± 0.70 ^d^	3.58 ± 0.74 ^c^	5999 ± 170 ^c^	4368 ± 119 ^c^	1290 ± 154 ^d^	5385 ± 142 ^d^
C4	2.30 ± 0.69 ^d^	4.64 ± 0.68 ^b^	5670 ± 132 ^d^	4368 ± 143 ^c^	1643 ± 187 ^d^	5712 ± 195 ^c^
C5	3.74 ± 0.40 ^a^	6.85 ± 0.94 ^a^	7168 ± 154 ^a^	5712 ± 159 ^a^	3772 ± 210 ^a^	7574 ± 178 ^a^
C6	2.43 ± 0.31 ^b^	3.19 ± 0.24 ^c^	6013 ± 118 ^c^	5040 ± 144 ^b^	2208 ± 121 ^c^	6052 ± 153 ^c^
C7	1.94 ± 0.11 ^d^	4.45 ± 0.79 ^b^	5971 ± 201 ^c^	5088 ± 147 ^b^	1723 ± 136 ^c^	7411 ± 141 ^a^
C8	1.53 ± 0.78 ^d^	3.89 ± 0.11 ^b^	6860 ± 138 ^b^	5088 ± 165 ^b^	2894 ± 140 ^b^	6585 ± 134 ^b^
ST3	1.80 ± 0.36 ^d^	5.93 ± 0.52 ^a^	5271 ± 197 ^d^	4848 ± 98 ^b^	1924 ± 155 ^c^	5731 ± 125 ^c^
ST4	2.61 ± 0.24 ^b^	4.03 ± 0.68 ^d^	6034 ± 140 ^c^	5088 ± 178 ^b^	3379 ± 167 ^b^	7411 ± 147 ^a^
ST5	3.29 ± 0.22 ^a^	6.32 ± 0.13 ^a^	7231 ± 111 ^a^	5856 ± 135 ^a^	4305 ± 141 ^a^	7680 ± 136 ^a^

The experiments were performed in triplicate (*n* = 3). The results are reported as means  ± SDs. Different letters (a–d) within columns are significantly different at *p*  <  0.05 according to the analysis by Duncan’s multiple-range test. The unfermented CCP served as the control group for comparison.

**Table 3 animals-15-02311-t003:** ANOVA of variable effects for optimization of pectin lyase (PL) production.

Source	SS	DF	Mean Squares	F-Value	Prob. > F
Model	69.80	14	4.99	17.05	<0.0001
X_1_	2.44	1	2.44	8.35	0.0136
X_2_	6.38	1	6.38	21.80	0.0005
X_3_	2.61	1	2.61	8.91	0.0114
X_4_	2.76	1	2.76	9.44	0.0097
X_1_^2^	22.01	1	22.01	75.26	<0.0001
X_2_^2^	35.80	1	35.80	122.39	<0.0001
X_3_^2^	2.69	1	2.69	9.19	0.0104
X_4_^2^	9.50	1	9.50	32.48	<0.0001
X_1_X_2_	0.017	1	0.017	0.057	0.8149
X_1_X_3_	1.11	1	1.11	3.78	0.0756
X_1_X_4_	0.026	1	0.026	0.088	0.7720
X_2_X_3_	4.30	1	4.30	14.71	0.0024
X_2_X_4_	4.80	1	4.80	16.42	0.0016
X_3_X_4_	0.054	1	0.054	0.19	0.6745
Residual	3.51	12	0.29		
Lack of fit	2.15	7	0.31	1.14	0.4603
Pure error	1.36	5	0.27		
Cor. total	73.31	26			
R^2^	0.9521				

**Table 4 animals-15-02311-t004:** ANOVA of variable effects for optimization of polygalacturonase (PG) production.

Source	SS	DF	Mean Squares	F-Value	Prob. > F
Model	59.87	14	4.28	12.94	<0.0001
X_1_	2.59	1	2.59	7.85	0.0160
X_2_	4.49	1	4.49	13.59	0.0031
X_3_	1.84	1	1.84	5.57	0.0360
X_4_	1.96	1	1.96	5.92	0.0316
X_1_^2^	19.26	1	19.26	58.31	<0.0001
X_2_^2^	32.27	1	32.27	97.67	<0.0001
X_3_^2^	1.79	1	1.79	5.41	0.0384
X_4_^2^	7.73	1	7.73	23.38	0.0004
X_1_X_2_	0.0003	1	0.0003	0.0009	0.9756
X_1_X_3_	1.18	1	1.18	3.58	0.0827
X_1_X_4_	0.015	1	0.05	0.047	0.8328
X_2_X_3_	3.42	1	3.42	10.36	0.0074
X_2_X_4_	3.35	1	3.35	10.15	0.0078
X_3_X_4_	0.081	1	0.081	0.25	0.6288
Residual	3.96	12	0.33		
Lack of fit	1.38	7	0.20	0.38	0.8787
Pure error	2.58	5	0.52		
Cor. total	63.83	26			
R^2^	0.9379				

**Table 5 animals-15-02311-t005:** Antioxidant profile and phenolic compounds of unfermented and fermented cherry coffee pulp powders.

	Unfermented CCP Powder	Fermented CCP Power
DPPH (µmole TE/L)	4290.82 ± 262 ^b^	8545.92 ± 241 ^a^
ABTS (µmole TE/L)	5573.33 ± 180 ^b^	10,495.56 ± 192 ^a^
FRAP (µmole TE/L)	5336.96 ± 287 ^b^	10,496.78 ± 176 ^a^
TPC (µmole GAE/L)	3236.11 ± 191 ^b^	8266.50 ± 154 ^a^
Gallic acid (µg/g)	102.46 ± 1.64 ^b^	215.83± 1.44 ^a^
Chlorogenic acid (µg/g)	344.07 ± 1.23 ^b^	512.35 ±1.56 ^a^
Caffeine (mg/g)	7.21 ± 0.85 ^b^	8.85 ± 1.02 ^a^
*p*-coumaric (µg/g)	5.92 ± 0.24 ^a^	3.64 ± 0.19 ^b^
Sinapic acid (µg/g)	10.25 ± 0.28 ^b^	22.88 ± 0.32 ^a^

The experiments are performed in triplicate (*n* = 3). The results are reported as means  ± SDs. Different letters (a,b) within rows are significantly different at *p*  <  0.05 according to the analysis by Duncan’s multiple-range test.

## Data Availability

The original contributions presented in this study are included in this article; further inquiries can be directed to the corresponding author.

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
