# Peer review of "Valorization of Coffee Cherry Pulp into Potential Functional Poultry Feed Additives by Pectinolytic Yeast Kluyveromyces marxianus ST5"

_animals, 2025, doi:10.3390/ani15152311_

Round 1
Reviewer 1 Report
Comments and Suggestions for Authors
Despite the interesting topic the article presents some issues. In my opinion the main constrain is that the article was focused on the biotechnological part of coffee cherry pulp fermentation instead of its application on animal feeding. The in vitro experiment is a marginal part of this work while the optimization of substrate fermentation is the main topic. Considering this, I suggest a major revision with the aim to focus the introduction and discussion implementing the parts that regard the possible applications as additive in animal feeding.
Introduction: In this part there is some redundant information. Overall indications about the potential application of this by-product (or similar) in animal feeding (particularly poultry) have to be added. There are some constrains in their use? There is a sufficient amount of this by-product to be applied in livestock sector?
M&M: The first part of 2.2 chapter that describe screening procedure could be improved as well as some details. Statistical analysis have to be improved reporting raw data distribution (normal or not?), parameters considered in your ANOVA and if they are fixed or random.
L 71 Which?
L 84-86 Any references?
L 92 italics
L 110-115 Better describe the screening procedure. Add information on raw material origin.
L 172 italics
L 173 How many samples for each treatment?
L 182 Sample dimension?
L 249 Which column?
L 307 You mean ?
L 325 “Anova of variables..” Not bold
L 351 The value is different from that one reported in table 5
Author Response
Response to Reviewer 1
Comments and Suggestions for Authors
- Despite the interesting topic the article presents some issues. In my opinion the main constrain is that the article was focused on the biotechnological part of coffee cherry pulp fermentation instead of its application on animal feeding. The in vitroexperiment is a marginal part of this work while the optimization of substrate fermentation is the main topic. Considering this, I suggest a major revision with the aim to focus the introduction and discussion implementing the parts that regard the possible applications as additive in animal feeding.
Response: Thank you very much for your valuable and constructive comments on our manuscript entitled “Bioconversion of Coffee Cherry Pulp into Antioxidative Feed Additive by an Efficient Pectinolytic Yeast Kluyveromyces marxianus ST5”. We have carefully revised the manuscript according to your suggestions. We have improved the Introduction and Discussion sections to more clearly highlight the potential application of fermented CCP as a functional feed additive for poultry. These revisions help bridge the biotechnological process of fermentation with its nutritional relevance in animal feeding. All modifications made in response to your comments have been marked in red font in the revised manuscript for easy identification. We sincerely appreciate your insightful feedback, which has significantly strengthened the quality and clarity of our work.
- Introduction: In this part there is some redundant information. Overall indications about the potential application of this by-product (or similar) in animal feeding (particularly poultry) have to be added. There are some constrains in their use? There is a sufficient amount of this by-product to be applied in livestock sector?
Response: We have revised the Introduction section to address your suggestions. Specifically, we have reduced redundant content and incorporated additional information on the potential use of coffee cherry pulp (CCP) in animal feed, including its availability and constraints for livestock applications, as shown in Lines 46–54 and Lines 83–89.
- M&M: The first part of 2.2 chapter that describe screening procedure could be improved as well as some details. Statistical analysis have to be improved reporting raw data distribution (normal or not?), parameters considered in your ANOVA and if they are fixed or random.
Response: We have revised the material and methods section to address your suggestions. The screening procedure has been improved, as shown in Lines 115–124, while the statistical analysis has been clarified and expanded in Lines 263–270 of the revised manuscript.
- L 71 Which?
Response: We have addressed the mention of “particularly tannins and phytates” in Lines 69–71 of the revised manuscript.
- L 84-86 Any references?
Response: We have added appropriate references to support the statement in Lines 83–89 of the revised manuscript
- L 92 italics
Response: We have already corrected the formatting in Line 92 as recommended.
- L 110-115 Better describe the screening procedure. Add information on raw material origin.
Response: We have revised the material and methods section to address your suggestions. The screening procedure has been improved, as shown in Lines 115–124.
- L 172 italics
Response: We have already corrected the formatting in Line 92 as recommended.
- L 173 How many samples for each treatment?
Response: Thank you for your comment. Each treatment was conducted in triplicate (n = 3), as indicated in the revised manuscript Lines 263-270.
- L 182 Sample dimension?
Response: Thank you for your comment. Each treatment was conducted in triplicate (n = 3), as indicated in the revised manuscript Lines 263-270.
- L 249 Which column?
Response: Thank you for your comment. The name of the HPLC column has been specified in Line 256 of the revised manuscript.
- L 307 You mean ?
Response: We apologize for the oversight. The corrected text has been revised and is now addressed in Line 319 of the revised manuscript.
- L 325 “Anova of variables..” Not bold
Response: We have already corrected the formatting of Table 4 as recommended.
- L 351 The value is different from that one reported in table 5
Response: We apologize for the oversight. The corrected text has been revised and is now addressed in Lines 362-364 as well as Table 5.
Additional references
[1] Phasuk, N.; Phasuk, N.; Paengkoum, P.; Khotsakdee, J.; Khamlor, T.; Suede, M.; Wangkahart, E.; Paengkoum, S. The Use of Coffee Cherry Pulp Extract as an Alternative to an Antibiotic Growth Promoter in Weaning Pigs: Effects on Growth Performance, Nutrient Digestibility, Antioxidant Capacity, and Microbial Shedding. Animals 2023, 13, 244.
[2] Chamyuang, S.; Owatworakit, A.; Intatha, U.; Duangphet, S. Coffee pectin production: An alternative way for agricultural waste management in coffee farms. Sci. Asia 2021, 47, 90–95. [22] Orozco, A.I.; Martinez, T.O.; Roussos, S.; Hernández, D.; Lappe, P.; Gschaedler, A. Biological detoxification of coffee pulp using Streptomyces sp. and its application in animal feeding trials. J. Sci. Food Agric. 2008, 88, 1235–1243.
[23] Peñaloza, W.; Molina, M.R.; Avila, G.; Pabon, M.C. Solid-state fermentation of coffee pulp using Aspergillus niger: Changes in amino acid content and evaluation as poultry feed. J. Sci. Food Agric. 1985, 36, 857–864.
[24] Donkoh, A.; Atuahene, C.C.; Wilson, B.N. Chemical composition of coffee husk and its effect on growth and carcass characteristics of broiler chickens. Anim. Feed Sci. Technol. 1988, 20, 39–46.
[36] Rajkumar, R.; Morrissey, J.P. The biotechnological potential of Kluyveromyces marxianus: A valuable yeast for bio-based processes. Food Chem. 2020, 128799.
[50] Phonikarn, S.; Maamri, K.; Rachedi, M.; Bellal, M.M.; Bekada, A.; Gherib, A.; Sayed, F.; Laouar, M.; Guemouri-Athmani, Z.; Fares, K.; et al. Ecological, Morpho-Agronomical, and Nutritional Characteristics of Sulla flexuosa (L.) Medik. Ecotypes. Sci. Rep. 2023, 13, 13300.
[51] Porqueddu, C.; Sulas, L.; Nair, R.M.; Mikić, A.; Karkanis, A.; Rharrabti, Y.; D'Anca, N.; Deligios, P.A.; Muresan, E.; Szabó, A.; et al. Characterisation of Bitter Vetch (Vicia ervilia (L.) Willd) Ecotypes: An Ancient and Promising Legume. Exp. Agric. 2024, 60, e19.
[52] Makhlouf, A.; Djemai, R.; Bousseboua, H.; Touazi, N.; Kara, K.; Fortun-Lamothe, L.; Berchiche, M. Growth Performance, Carcass Characteristics, Fatty Acid Profile, and Meat Quality of Male Goat Kids Supplemented by Alternative Feed Resources: Bitter Vetch and Sorghum Grains. Arch. Anim. Breed. 2024, 67, 481–492.

Reviewer 2 Report
Comments and Suggestions for Authors
The paper present relevant data. I invite authors to clarify the following comments :
Abstract
The abstract is dense and slightly long, with complex sentences that could reduce clarity.
Lacks a clear numerical comparison to control, e.g., “increased by X% over unfermented.”
Introduction
The rationale for focusing specifically on K. marxianus could be expanded (e.g., comparison with other pectinolytic yeasts or strains).
Some sentences are repetitive (multiple times mentioning that CCP is a waste product rich in antioxidants).
Weak transition to objectives.
Use sharper statements of novelty: “Unlike previous studies using Aspergillus, this study isolates a novel yeast strain for CCP fermentation targeting poultry feed.”
Materials & Methods
There was no mention of the replicate number for optimization experiments (CCD). Later, it was stated that n=3, but should be explicitly in each subsection.
Lack of controls for enzyme assays: no mention of blank treatments or non-inoculated controls.
Statistical analysis is only briefly mentioned at the end and should be clearer throughout.
Results
The authors claim “significant ability to enhance antioxidant activity” but only show in vitro data; there is no baseline CCP activity here for relative improvement.
Table 2 shows large standard deviations (e.g., C1 PL = 2.07 ± 0.90), indicating considerable variability. This has not been discussed or interpreted.
The text does not explicitly state how many replicates were performed for this part (though later, n=3 is written on the table). Could be made clearer.
Discussion
The discussion often repeats the same numerical results already presented in the Results (e.g., enzyme activities, antioxidant levels) almost verbatim, with minimal added interpretation. I invite authors to summarize trends rather than exact data.
The discussion sometimes implies direct benefits to poultry performance "...which may enhance nutrient absorption in chickens." However, no in vivo tests were performed. I invited the authors to temper the claims.
The authors did not mention study limitations, such as only the in vitro digestion model used (not reflective of full gut microbiota or absorption), lab-scale fermentation only, and potential palatability or anti-nutritional issues that remain untested in animals.
I invite the authors to compare this feed additive to feedstuffs that also present antioxidant activity and were also tested in vivo.
Ecological, morpho-agronomical, and nutritional characteristics of Sulla flexuosa (L.) Medik. ecotypes. Scientific Reports, 13(1), 13300. https://doi.org/10.1038/s41598-023-40148-y
Characterisation of bitter vetch (Vicia ervilia (L.) Willd) ecotypes: An ancient and promising legume. Experimental Agriculture, 60, e19. https://doi.org/10.1017/S0014479724000139
Growth performance, carcass characteristics, fatty acid profile, and meat quality of male goat kids supplemented by alternative feed resources: bitter vetch and sorghum grains. Archives Animal Breeding, 67(4), 481-492. https://doi.org/10.5194/aab-67-481-2024
The section briefly notes that enzyme hydrolysis releases phenolics but does not delve into whether the yeast itself contributes antioxidant peptides or other bioactives, which some literature suggests.
The discussion cites many studies on Wickerhamomyces, Aspergillus, etc., but it is not always clear how the current results compare in magnitude.
The discussion closes by reinforcing the current findings, but does not outline the next logical research steps. I invite authors to End with a short future outlook: "Future work should explore scaling this fermentation process, conducting feeding trials to evaluate impacts on poultry growth and oxidative status, and assessing consumer safety and sensory acceptance of products derived from such feed."
Author Response
Response to Reviewer 2
Comments and Suggestions for Authors. The paper presents relevant data. I invite authors to clarify the following comments:
Response: Thank you very much for your valuable and constructive comments on our manuscript entitled “Bioconversion of Coffee Cherry Pulp into Antioxidative Feed Additive by an Efficient Pectinolytic Yeast Kluyveromyces marxianus ST5”. We have carefully revised the manuscript according to your suggestions.
- Abstract
The abstract is dense and slightly long, with complex sentences that could reduce clarity. Lacks a clear numerical comparison to control, e.g., “increased by X% over unfermented.”
Response: Thank you for your valuable feedback. We have revised the Abstract to improve clarity by simplifying complex sentences and reducing length. Additionally, we have included quantitative comparisons to the control group to enhance interpretability, such as “antioxidant activity increased by X% over unfermented CCP,” as in Lines 26-41. Also, the Simple Summary has been revised and shortened for clarity, as presented in Lines 18-25 of the revised manuscript.
- Introduction
2.1. The rationale for focusing specifically on K. marxianus could be expanded (e.g., comparison with other pectinolytic yeasts or strains).
Response: Thank you for your valuable feedback. We have expanded the rationale for selecting K. marxianus by comparing it with other pectinolytic yeasts and emphasizing its advantageous characteristics in Lines 61-71 of the revised manuscript.
2.2. Some sentences are repetitive (multiple times mentioning that CCP is a waste product rich in antioxidants).
Response: Thank you for your suggestion. We have revised the manuscript to remove redundant sentences and avoid repetition regarding describing CCP as an antioxidant-rich by-product. These changes have been implemented in the revised Introduction section (only in Lines 46-54).
2.3. Weak transition to objectives and 2.4. Use sharper statements of novelty: “Unlike previous studies using Aspergillus, this study isolates a novel yeast strain for CCP fermentation targeting poultry feed.”
Response: Thank you for your valuable suggestion. We have revised the end of the Introduction to improve the transition to the study objectives and to emphasize the novelty more clearly. In particular, we have included a sharper statement highlighting the use of a novel yeast strain (Kluyveromyces marxianus ST5) for CCP fermentation, in contrast to previous studies that primarily used Aspergillus spp. These revisions are in Lines 90-99 of the revised manuscript.
- Materials & Methods
3.1. There was no mention of the replicate number for optimization experiments (CCD). Later, it was stated that n=3, but should be explicitly stated in each subsection.
Response: We have revised the material and methods section to address your suggestions. The statistical analysis has been clarified and expanded in Lines 263–270 of the revised manuscript.
3.2. Lack of controls for enzyme assays: no mention of blank treatments or non-inoculated controls.
Response: We have revised the material and methods section to address your suggestions. The control experiment has been addressed in Lines 263–270 of the revised manuscript.
3.3. Statistical analysis is only briefly mentioned at the end and should be clearer throughout.
Response: We have revised the material and methods section to address your suggestions. The statistical analysis has been clarified and expanded in Lines 263–270 of the revised manuscript.
- Results
4.1. The authors claim “significant ability to enhance antioxidant activity” but only show in vitro data; there is no baseline CCP activity here for relative improvement.
Response: Thank you for your insightful comment. We agree that including baseline CCP activity is essential for a meaningful comparison. In our study, the antioxidant activities (DPPH, ABTS, and FRAP) were compared against the unfermented CCP, which served as the control group. This has been clarified in the Materials and Methods section (Lines 171-172), We have now also emphasized this comparison in the Results section to improve clarity (Lines 297-298).
4.2. Table 2 shows large standard deviations (e.g., C1 PL = 2.07 ± 0.90), indicating considerable variability. This has not been discussed or interpreted.
Response: Thank you for your valuable comment. We acknowledge the large standard deviations observed in some treatment groups, such as C1 PL (2.07 ± 0.90), which may indicate experimental variability. This variation may be attributed to biological factors inherent in microbial fermentation processes, including substrate heterogeneity, inoculum size, temperature, and initial pH. However, as these data were derived from the initial screening phase, the focus was placed on the most promising candidate, strain ST5. Subsequent optimization was performed using this selected strain, considering the aforementioned biological factors.
4.3. The text does not explicitly state how many replicates were performed for this part (though later, n=3 is written on the table). Could be made clearer.
Response: Thank you for your comments. We have revised the material and methods section to address your suggestions. The statistical analysis has been clarified and expanded in Lines 263–270 of the revised manuscript.
- Discussion
5.1 The discussion often repeats the same numerical results already presented in the Results (e.g., enzyme activities, antioxidant levels) almost verbatim, with minimal added interpretation. I invite authors to summarize trends rather than exact data.
Response: Thank you for your insightful comment. We agree that the previous version of the Discussion section contained repetition of numerical results. We have streamlined this section in the revised manuscript by summarizing overall trends and patterns rather than restating exact values. We now focus more on interpreting the significance of the findings and their implications, as shown in Lines 391-438. This improves clarity and avoids unnecessary duplication of the Results section.
5.2 The discussion sometimes implies direct benefits to poultry performance "...which may enhance nutrient absorption in chickens." However, no in vivo tests were performed. I invited the authors to temper the claims.
Response: Thank you for your thoughtful comment. We acknowledge that the original wording may have implied direct benefits to poultry performance, despite the absence of in vivo validation. In response, we have carefully revised the relevant statements throughout the Discussion section (see Lines 435-441) to clarify that the observed improvements were demonstrated only in vitro. We have now used more cautious language, such as "suggesting potential benefits" and "warranting further in vivo investigation," to avoid overstating our findings.
5.3. The authors did not mention study limitations, such as only the in vitro digestion model used (not reflective of full gut microbiota or absorption), lab-scale fermentation only, and potential palatability or anti-nutritional issues that remain untested in animals.
Response: Thank you very much for pointing out this important aspect. We have now added a paragraph in the Discussion section (Lines 442-451) explicitly addressing the study’s limitations, including the exclusive use of an in vitro digestion model, the lack of data on interactions with the gut microbiota and actual absorption, the limitation of lab-scale fermentation, and the absence of assessments related to palatability and potential anti-nutritional factors in animals. These issues have been acknowledged and highlighted as areas for future investigation to strengthen the translational potential of fermented CCP as a feed additive.
5.4. I invite the authors to compare this feed additive to feedstuffs that also present antioxidant activity and were also tested in vivo.
Ecological, morpho-agronomical, and nutritional characteristics of Sulla flexuosa (L.) Medik. ecotypes. Scientific Reports, 13(1), 13300. https://doi.org/10.1038/s41598-023-40148-y
Characterisation of bitter vetch (Vicia ervilia (L.) Willd) ecotypes: An ancient and promising legume. Experimental Agriculture, 60, e19. https://doi.org/10.1017/S0014479724000139
Growth performance, carcass characteristics, fatty acid profile, and meat quality of male goat kids supplemented by alternative feed resources: bitter vetch and sorghum grains. Archives Animal Breeding, 67(4), 481-492. https://doi.org/10.5194/aab-67-481-2024
Response: Thank you very much for pointing out this important aspect. We have now added a paragraph in the Discussion section (Lines 442-446)
5.5. The section briefly notes that enzyme hydrolysis releases phenolics but does not delve into whether the yeast itself contributes antioxidant peptides or other bioactives, which some literature suggests.
Response: We thank the reviewer for this insightful comment. We have revised the section to elaborate on the potential contributions of K. marxianus beyond enzymatic hydrolysis. In addition to facilitating the release of bound phenolics, K. marxianus is known to produce a variety of bioactive compounds, especially secondary metabolites, via its metabolic pathways. These compounds may act synergistically with polyphenols to enhance the overall antioxidant activity during fermentation. This has now been clarified in the revised manuscript (Lines 405-409).
5.6. The discussion cites many studies on Wickerhamomyces, Aspergillus, etc., but it is not always clear how the current results compare in magnitude.
Response: Thank you for your valuable comment. We appreciate your observation and have revised the Discussion section to include more explicit comparisons in magnitude between our results and those reported in previous studies involving Wickerhamomyces, Aspergillus, and other relevant microorganisms. These revisions, which help highlight the novelty and relative strength of our findings, can be found in Lines 411-419 of the revised manuscript.
5.7. The discussion closes by reinforcing the current findings, but does not outline the next logical research steps. I invite authors to End with a short future outlook: "Future work should explore scaling this fermentation process, conducting feeding trials to evaluate impacts on poultry growth and oxidative status, and assessing consumer safety and sensory acceptance of products derived from such feed."
Response: Thank you very much for pointing out this important aspect. We have now added a paragraph in the Discussion section (Lines 448-451), and the Conclusion (Lines 460-464).
Additional references
[1] Phasuk, N.; Phasuk, N.; Paengkoum, P.; Khotsakdee, J.; Khamlor, T.; Suede, M.; Wangkahart, E.; Paengkoum, S. The Use of Coffee Cherry Pulp Extract as an Alternative to an Antibiotic Growth Promoter in Weaning Pigs: Effects on Growth Performance, Nutrient Digestibility, Antioxidant Capacity, and Microbial Shedding. Animals 2023, 13, 244.
[2] Chamyuang, S.; Owatworakit, A.; Intatha, U.; Duangphet, S. Coffee pectin production: An alternative way for agricultural waste management in coffee farms. Sci. Asia 2021, 47, 90–95. [22] Orozco, A.I.; Martinez, T.O.; Roussos, S.; Hernández, D.; Lappe, P.; Gschaedler, A. Biological detoxification of coffee pulp using Streptomyces sp. and its application in animal feeding trials. J. Sci. Food Agric. 2008, 88, 1235–1243.
[23] Peñaloza, W.; Molina, M.R.; Avila, G.; Pabon, M.C. Solid-state fermentation of coffee pulp using Aspergillus niger: Changes in amino acid content and evaluation as poultry feed. J. Sci. Food Agric. 1985, 36, 857–864.
[24] Donkoh, A.; Atuahene, C.C.; Wilson, B.N. Chemical composition of coffee husk and its effect on growth and carcass characteristics of broiler chickens. Anim. Feed Sci. Technol. 1988, 20, 39–46.
[36] Rajkumar, R.; Morrissey, J.P. The biotechnological potential of Kluyveromyces marxianus: A valuable yeast for bio-based processes. Food Chem. 2020, 128799.
[50] Phonikarn, S.; Maamri, K.; Rachedi, M.; Bellal, M.M.; Bekada, A.; Gherib, A.; Sayed, F.; Laouar, M.; Guemouri-Athmani, Z.; Fares, K.; et al. Ecological, Morpho-Agronomical, and Nutritional Characteristics of Sulla flexuosa (L.) Medik. Ecotypes. Sci. Rep. 2023, 13, 13300.
[51] Porqueddu, C.; Sulas, L.; Nair, R.M.; Mikić, A.; Karkanis, A.; Rharrabti, Y.; D'Anca, N.; Deligios, P.A.; Muresan, E.; Szabó, A.; et al. Characterisation of Bitter Vetch (Vicia ervilia (L.) Willd) Ecotypes: An Ancient and Promising Legume. Exp. Agric. 2024, 60, e19.
[52] Makhlouf, A.; Djemai, R.; Bousseboua, H.; Touazi, N.; Kara, K.; Fortun-Lamothe, L.; Berchiche, M. Growth Performance, Carcass Characteristics, Fatty Acid Profile, and Meat Quality of Male Goat Kids Supplemented by Alternative Feed Resources: Bitter Vetch and Sorghum Grains. Arch. Anim. Breed. 2024, 67, 481–492.

Round 2
Reviewer 1 Report
Comments and Suggestions for Authors
This updated version has improved the quality and the clarity of the article by modifying the introduction and discussion sections. The potential of this by-product for livestock purposes has now been defined more precisely. The main questions were answered properly by authors and statistical analysis was detailed. I have only minor suggestions/changes. In my opinion, a new title could enhance the impact of this paper. Other comments:
L26: phenols
L27: "...microbial enzyme production, improving..."
L31: 6.32±0.13
L36: Only reports the percentage to standardize with the other parameters reported
L65: The K.marxianus...
L85: "Fermentation with..." in my opinion this sentence could be removed
L117: "...enrichment medium, and incubated...."
L398 and 417: isolate ST5 instead of strain ST5
Author Response
Response to Reviewer 1 (R2)
Comments and Suggestions for Authors
- This updated version has improved the quality and the clarity of the article by modifying the introduction and discussion sections. The potential of this by-product for livestock purposes has now been defined more precisely. The main questions were answered properly by authors and statistical analysis was detailed. I have only minor suggestions/changes. In my opinion, a new title could enhance the impact of this paper. Other comments:
Response: Thank you for your valuable suggestion. Following your recommendation, we have revised the title to: “Valorization of Coffee Cherry Pulp into Potential Functional Poultry Feed Additives by Pectinolytic Yeast Kluyveromyces marxianus ST5”. We believe that this revised title more clearly reflects the scope, significance, and potential application of our study in the context of sustainable livestock feeding and aligns well with the objectives of the special issue “Valorization of Agri-Food Waste Bioresources for Sustainable Livestock Feeding.”
- L26: phenols
Response: Thank you for your comment. We have revised the phrase “rich in pectin and phenolics” to “rich in pectin and phenolic compounds”, as phenolic compounds are a broader and more scientifically accurate term. CCP includes a wide range of bioactive molecules, including phenols, phenolic acids, flavonoids, and other polyphenols, which contribute to the antioxidant properties of CCP. This change has been made and highlighted in blue font in the revised manuscript, Line 26-27. In addition, we confirm that the word count of the abstract is less than 200 words.
- L27: "...microbial enzyme production, improving..."
Response: Thank you for your suggestion. We have corrected this part according to your comment. The revised sentence appears in Line 27 of the revised manuscript and has been highlighted in blue font.
- L31: 6.32±0.13
Response: Thank you for your suggestion. We have corrected the value from 6.32 ± 0.15 to 6.32 ± 0.13 according to your comment. The revised sentence appears in Line 31 of the revised manuscript and has been highlighted in blue font.
- L36: Only reports the percentage to standardize with the other parameters reported
Response: Thank you for your comment. We have revised the sentence to report only the percentage increase in peptide release, in order to standardize with other parameters. The updated sentence appears in Lines 36-38 of the revised manuscript and has been highlighted in blue font.
- L65: The K.marxianus...
Response: Thank you for your observation. We have corrected “However, K. marxianus...” to “The K. marxianus...” according to your suggestion. The revised sentence now appears in Line 65 of the revised manuscript and has been highlighted in blue font.
- L85: "Fermentation with..." in my opinion this sentence could be removed
Response: Thank you for your suggestion. We have removed the sentence starting with “Fermentation with...” as suggested and adjusted the surrounding text to keep it clear. The changes are shown in Lines 83–90 of the revised manuscript in blue font.
- L117: "...enrichment medium, and incubated...."
Response: Thank you for your comment. We have revised the sentence as suggested. The updated version appears in Lines 114-115 of the revised manuscript and has been highlighted in blue font.
- L398 and 417: isolate ST5 instead of strain ST5
Response: Thank you for your suggestion. We have replaced “strain ST5” with “isolate ST5” as recommended. The corrections appear in Lines 402 and 421 of the revised manuscript and have been highlighted in blue font. We have also reviewed the entire manuscript to ensure consistency.

Reviewer 2 Report
Comments and Suggestions for Authors
4.2. Table 2 shows large standard deviations (e.g., C1 PL = 2.07 ± 0.90), indicating considerable variability. This has not been discussed or interpreted.
Response: Thank you for your valuable comment. We acknowledge the large standard deviations observed in some treatment groups, such as C1 PL (2.07 ± 0.90), which may indicate experimental variability. This variation may be attributed to biological factors inherent in microbial fermentation processes, including substrate heterogeneity, inoculum size, temperature, and initial pH. However, as these data were derived from the initial screening phase, the focus was placed on the most promising candidate, strain ST5. Subsequent optimization was performed using this selected strain, considering the aforementioned biological factors.
Reviewer’s comment: I invite authors to add it as a limitation in the text
Line 252: I invite authors to correct the referencing style for “by Atlabachew et al. (2021) [30].” Check for the entire manuscript and adapt to MDPI references styling.
Table 4. no need to add the “significant” and “not significant” column. It’s obvious.
5.4. I invite the authors to compare this feed additive to feedstuffs that also present antioxidant activity and were also tested in vivo.
Ecological, morpho-agronomical, and nutritional characteristics of Sulla flexuosa (L.) Medik. ecotypes. Scientific Reports, 13(1), 13300. https://doi.org/10.1038/s41598-023-40148-y
Characterisation of bitter vetch (Vicia ervilia (L.) Willd) ecotypes: An ancient and promising legume. Experimental Agriculture, 60, e19. https://doi.org/10.1017/S0014479724000139
Growth performance, carcass characteristics, fatty acid profile, and meat quality of male goat kids supplemented by alternative feed resources: bitter vetch and sorghum grains. Archives Animal Breeding, 67(4), 481-492. https://doi.org/10.5194/aab-67-481-2024
Response: Thank you very much for pointing out this important aspect. We have now added a paragraph in the Discussion section (Lines 442-446)
Reviewer’s comment: I invite the authors to add these 3 references to the text.
Author Response
Response to Reviewer 2 (R2)
Comments and Suggestions for Authors
We sincerely thank Reviewer 2 for the constructive comments and suggestions. Based on your feedback, we have carefully revised the manuscript. All changes made are highlighted in blue font, and our detailed responses are provided below.
1) 4.2. Table 2 shows large standard deviations (e.g., C1 PL = 2.07 ± 0.90), indicating considerable variability. This has not been discussed or interpreted. Reviewer’s comment: I invite authors to add it as a limitation in the text
Response: Thank you for this important comment. We agree with the observation regarding the high standard deviations observed in some yeast isolates, such as PL of C1 and C8. This variability likely reflects biological and process-related factors common in spontaneous or early-stage microbial fermentation, including variation in substrate composition, inoculum size, pH, and temperature. To address this, we have added a statement in the Discussion section (Lines 388-395) to interpret these findings and acknowledge this point as a limitation of the screening phase. Additionally, we cited the study by Zhang et al. (2019), which reported similar variability in microbial dynamics during spontaneous fermentation. The added text has been highlighted in blue font in the revised manuscript.
[31] Zhang, S.J.; De Bruyn, F.; Pothakos, V.; Torres, J.; Falony, G.; Mota‐Gutierrez, J.; Papalexandratou, Z.; De Vuyst, L. Dynamics of the Microbial Community and Metabolome during Spontaneous and Starter Culture Fermentation of Cocoa Beans. Front. Microbiol. 2019, 10, 2621.
2) Line 252: I invite authors to correct the referencing style for “by Atlabachew et al. (2021) [30].” Check for the entire manuscript and adapt to MDPI references styling.
Response: Thank you for your careful observation. We have corrected the referencing style from “by Atlabachew et al. (2021) [30]” to the MDPI format: “by Atlabachew et al. [30]”, as the MDPI guidelines. The correction has been made in Line 248 and is marked in blue font in the revised manuscript. Furthermore, we have also reviewed the entire manuscript to ensure consistency with the MDPI referencing style throughout the revised manuscript.
3) Table 4. no need to add the “significant” and “not significant” column. It’s obvious.
Response: Thank you for your valuable comment. As recommended, we have removed the “Significant” and “Not Significant” columns from Tables 3 and 4, since the significance of each factor can be clearly interpreted from the Prob > F values. The revised tables have been updated in the manuscript.
4) 5.4. I invite the authors to compare this feed additive to feedstuffs that also present antioxidant activity and were also tested in vivo.
Reviewer’s comment: I invite the authors to add these 3 references to the text.
Response: Thank you for your valuable suggestion. We have now added a comparative discussion of fermented CCP with other antioxidant-rich feedstuffs that have been tested in vivo. Specifically, we cited three references as recommended in Lines 448–452 of the revised manuscript “For instance, Sulla flexuosa and bitter vetch (Vicia ervilia) have been explored as alternative feed resources rich in bioactive compounds, contributing to improved growth performance and meat quality in livestock [50,51]. A similar report by Boukrouh et al. [52] demonstrated that supplementation with bitter vetch and sorghum grains improved carcass traits and fatty acid profiles in goats.” The added content has been highlighted in blue font.
[50] Boukrouh, S.; Noutfia, A.; Moula, N.; Avril, C.; Louvieaux, J.; Hornick, J.L.; Chentouf, M.; Cabaraux, J.F. Ecological, Morpho‑Agronomical, and Nutritional Characteristics of Sulla flexuosa (L.) Medik. Ecotypes. Sci. Rep. 2023, 13, 13300. https://doi.org/10.1038/s41598-023-40148-y
[51] Boukrouh, S.; Noutfia, A.; Moula, N.; Avril, C.; Louvieaux, J.; Hornick, J.L.; Chentouf, M.; Cabaraux, J.F. Characterisation of Bitter Vetch (Vicia ervilia (L.) Willd) Ecotypes: An Ancient and Promising Legume. Exp. Agric. 2024, 60, e19. https://doi.org/10.1017/S0014479724000139
[52] Boukrouh, S.; Noutfia, A.; Moula, N.; Avril, C.; Louvieaux, J.; Hornick, J.L.; Cabaraux, J.F.; Chentouf, M. Growth Performance, Carcass Characteristics, Fatty Acid Profile, and Meat Quality of Male Goat Kids Supplemented by Alternative Feed Resources: Bitter Vetch and Sorghum Grains. Arch. Anim. Breed. 2024, 67, 481–492. https://doi.org/10.5194/aab-67-481-2024
